# CLOSING THE GAP ON TABULAR DATA WITH FOURIER AND IMPLICIT CATEGORICAL FEATURES

## ABSTRACT

While Deep Learning has demonstrated impressive results in applications on various data types, it continues to lag behind tree-based methods when applied to tabular data, often referred to as the last "unconquered castle" for neural networks. We hypothesize that a significant advantage of tree-based methods lies in their intrinsic capability to model and exploit non-linear interactions induced by features with categorical characteristics. In contrast, neural-based methods exhibit biases toward uniform numerical processing of features and smooth solutions, making it challenging for them to effectively leverage such patterns. We address this performance gap by using statistical-based feature processing techniques to identify features that are strongly correlated with the target once discretized. We further mitigate the bias of deep models for overly-smooth solutions, a bias that does not align with the inherent properties of the data, using Learned Fourier Features. We show that our proposed feature preprocessing significantly boosts the performance of deep learning models and enables them to achieve a performance that closely matches or surpasses XGBOOST on a comprehensive tabular data benchmark.

## 1 INTRODUCTION

Deep learning has demonstrated significant success in domains like natural language processing and image analysis. However, on tabular data, deep learning (DL) methods have not yet made a breakthrough, with methods based on decision tree (DT), such as eXtreme Gradient Boosting (XGBOOST) (Chen & Guestrin, 2016), still being superior. Tabular datasets present some interesting peculiarities relevant for the design of a learning system: it usually involves a small sample size, data often lies in a "natural base" (Ng, 2004) and the function from features to target variables can be highly non-smooth (Grinsztajn et al., 2022).

Furthermore tabular data collection often involves manual annotation of data types in numerical and categorical features. However, even if numerical in nature, some features might still be *implicitly categorical*. In such cases proper modelling assumptions are required in order to capture the possibly intricate and highly discontinuous interactions of the features with the target variable when using neural networks. For example, the `eye_movements` dataset (Salojärvi et al., 2005) measures the correlation between eye movements and relevant content in the context of information retrieval. Subjects were presented with multiple assignments, where each assignment contains a query and a list containing one correct sentence and several irrelevant or relevant sentences to the query. Even though these features are numerical in nature, the assignment number, line number and word number are features with categorical characteristics and properly encoding them leads to significant performance increases for methods.

An emerging line of work attempts to address the performance gap between DL and tree-based methods on tabular data, with an increasing number of deep methods claiming to surpass tree-based methods on some datasets. Recently, the focus has been shifting towards understanding the causes of the performance gap and to analysing the dataset intricacies that favors some models over the others.

In this vein, Grinsztajn et al. (2022) identifies robustness to uninformative features, preservation of the original data orientation and the ability to accommodate irregular functions as some of the main advantages of tree-based methods in tabular data learning. Additionally, the authors conclude that categorical variables are a minor weakness of neural networks.

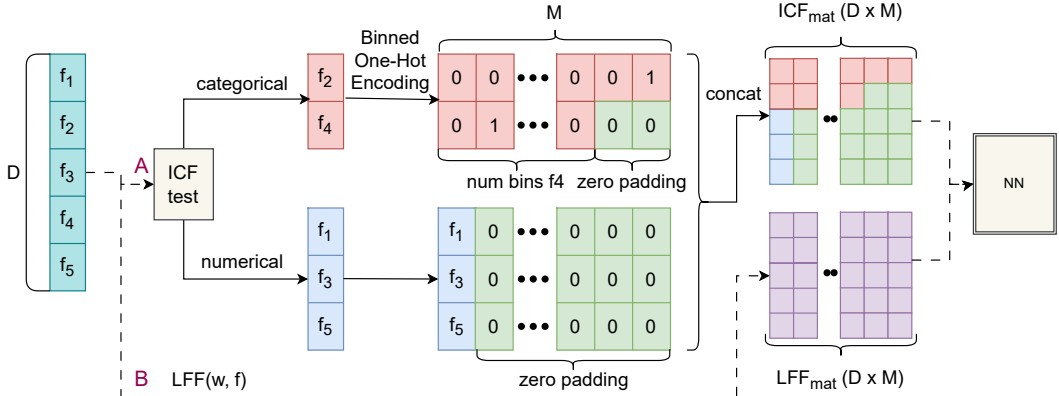

Figure 1: Method overview. We select either (A) ICF or (B) LFF for data preprocessing in each run. In ICF, we add zero-padding for features with a lower number of bins to allow concatenation.

In contrast, we highlight the heterogeneity inherent in tabular data and the brittleness of neural networks to numerical features with categorical characteristics. We coin these implicitly categorical features and demonstrate their substantial influence on the performance gap.

Inspired by the finding of Grinsztajn et al. (2022) regarding the over-smoothed solutions of DL we adapt Learned Fourier Features (LFF) for overcoming the smoothness bias (Tancik et al., 2020; Li & Pathak, 2021; Yang et al., 2022) of deep neural networks, leading to an important increase in performance when learning from tabular data applications.

Our **main contributions** can be summarized as follows:

1. Highlight the prevalence of numerical features with implicit categorical characteristics and their potentially significant impact on the design choices of deep learning (DL) models for tabular data.

2. Introduce several methods for identifying implicitly categorical features and demonstrate that a proper encoding is necessary for closing the performance gap towards decision tree (DT) models and, in some cases, leads to large performance "spikes" not reached by any other model in our comparison.

3. Our proposed Categorical Feature Detection (CFD), together with Learned Fourier Features (LFF) significantly improve the performance of deep learning methods on 68[1] tabular datasets when considering the result of a large hyper-parameter random search.

## 2 RELATED WORK

As revealed in the detailed survey of Borisov et al. (2022), a main line of work on tabular DL data focuses on data representations and encoding Yoon et al. (2020); Hancock & Khoshgoftaar (2020); Bahri et al. (2021); Sun et al. (2019), tuning deep neural networks through high regularization Kadra et al. (2021); Shavitt & Segal (2018) and specialized architectures: hybrid Ke et al. (2019); Luo et al. (2021); Ivanov & Prokhorenkova (2021); Luo et al. (2021); Popov et al. (2019) and transformers Somepalli et al. (2021); Arik & Pfister (2021); Huang et al. (2020). A comparison between gradient boosting methods and DL for tabular data detailed in Shwartz-Ziv & Armon (2022) shows that, despite of the superiority of gradient boosting methods, their ensemble with DL methods outperforms them. Other hybrid methods and their performance are further highlighted in Sarkar (2022).

Our work is closely related with Gorishniy et al. (2022), which focuses on the impact of proper embedding schemes on the performance of deep learning models on DT-favored datasets. This recent

---

[1]The benchmark by Grinsztajn et al. (2022) has a total of 69 tasks. We leave out from our reporting tasks in which no model achieves a score higher than 0.1, which leaves us with 68. Details in Section 5

work focuses on embeddings based on piecewise linear encoding and periodic activation functions to improve both multi-layer Perceptron (MLP) and transformer-based architectures' performance on tabular data, stacking the embedding types in different combinations. Another recent work, Hollmann et al. (2022), introduces a new transformer architecture trained to approximate Bayesian inference on synthetic datasets drawn from a prior. A different approach, based on modern Hopfield networks, is introduced in Schäfl et al. (2022). Joseph & Raj (2022) proposes a similar approach with Popov et al. (2019) with elements inspired from the Gated recurrent unit. Furthermore, the performance of DL methods is shown to be substantially improved by pre-training on upstream data to increase performance on the target dataset in Levin et al. (2022).

Another emerging direction focuses on studying the the gap between tree-based models and DL and the particularities of behaviour of these approaches when generalizing on tabular data. Specifically, Grinsztajn et al. (2022) conducts an extensive empirical analysis of the gap between these two clsases of models on a benchmark of 45 tabular datasets, observing robustness to uninformative features, non-invariance to orientation changes and bias towards non-smooth decision boundaries as desired properties for a model's success on tabular data. Furthermore, McElfresh et al. (2023) conducts a large scale tabular data analysis on 176 datasets and studying the trends between the best performing model class and a large set of metafeatures, such as dataset irregularity, ratio of the dataset size to number of features and target class frequencies. The experiments reveal that the performance gap is negligible for a surprisingly large number of datasets.

## 3 PROPOSED METHODS

We aim to identify implicitly categorical features using simple statistical methods that quantify the correlation between a given feature and the target for a categorical encoding of the feature. Intuitively, implicitly categorical features will exhibit statistical significant correlations with the target when the feature is categorized such that its notion of numerical distance is removed. All the implicitly categorical features identification tests are applied on the training data. We address the bias of deep methods towards overly-smooth solutions by using Learned Fourier Features. We present an overview of our method in Figure 1, describe the implementation details in 3.2 and 3.3 and present an algorithmic description of our feature preprocessing in Appendix A.3. We use our preprocessing method on top of two backbone models with different particularities in regard to rotational variance: MLP and 1D convolutional Residual Network (RESNET), detailed in 3.4.

### 3.1 CATEGORICAL ENCODING

We one-hot encode the categorical features and transpose them along channels. Consider a sample $\boldsymbol{X}^D$, where $D$ is the number of features or columns of the sample. Each scalar element $\boldsymbol{X}_i$ can be transformed either into a categorical or numerical feature. Categorical encoding is performed by concatenating the (possibly binarised) one-hot-encoding with the original value of the feature: $\Phi_i^M = \text{OHE}(X_i)$, where $M$ corresponds to the largest number of categories across features of $X$. Numerical encoding is then simply the zero-padded vector of size $M$, with the original feature value at the first position $\Phi_i^M = [X_i, \boldsymbol{0}^{M-1}]$. Therefore, we use an input of size $D$ with $M$ channels for the RESNET model as it supports multi-channel data, while the input for MLP is flattened.

Selecting the features of $\boldsymbol{X}$ for which we apply the categorical encoding is done by identifying the indices of the *implicitly categorical features* first. We name the procedure through which we identify the relevant features and then apply the categorical encoding CFD and the resulting model, **ResNet+C** or **MLP+C**. Next, we propose and describe several simple statistical methods for the identification of implicitly categorical features.

### 3.2 IMPLICITLY CATEGORICAL FEATURE IDENTIFICATION

We postulate the presence of numerical features that exhibit statistically significant correlations with the feature space and/or the target value in a categorical manner, which we term implicitly categorical features. We propose a method of identifying implicitly categorical features using basic statistical methods. The identified features are subsequently binned and encoded as described in section 3.1. We make the assumption that implicitly categorical features have a low cardinality and consequently, we only perform statistical tests on columns with less than $5000$ unique values (details

in Appendix A.2). Additionally, given the sensitivity of some of the statistical tests to low cardinality features, we use a hyperparameter that decides whether or not features with a low cardinality should be automatically considered as implicitly categorical.

We choose to identify implicitly categorical features with several statistical tests. Chi-squared usually tests the hypothesis that two categorical variables are independent. By categorising a numerical feature through binning, we test whether or not the feature and the target in the classification tasks are correlated. If the test obtains a low p-value (which measures the evidence against a null hypothesis that the two categorical features are correlated), we encode the feature as categorical. By properly adjusting the threshold, we can filter out features that don't exhibit significant correlations with the target when discretized. Additionally, we use the ANOVA test for regression or the Mutual Info, which is a metric that also accounts for one feature's correlation with the other features in the dataset and not only the target.

### 3.2.1 CLASSIFICATION TASKS

For the classification datasets, we use the $\chi^2$ (chi-square) hypothesis test Pearson (1900) and consider the features with a p-value below a threshold as categorical, as shown in Equation 1. Additionally, we consider features with a low cardinality as categorical.

$$\text{ICF}(\boldsymbol{X}) = \{i \mid \text{p\_value}(\chi^2(\boldsymbol{X}_i, Y)) < \chi^2_{thresh}\} \tag{1}$$

### 3.2.2 REGRESSION TASKS

For regression datasets, we select categorical-behaved features using the one-way Analysis of Variance (ANOVA) Girden (1992) test and Mutual Info Thomas & Cover (1991) test.

For the ANOVA test, we select the features with a p_value below a threshold for the statistical test of groups of $Y$, where the group are selected as the values of $Y$ for each possible value for the feature i, as depicted in Equation 2, where $Y_{vik}$ refers to values of $Y$ for the data samples where the value of $\boldsymbol{X}_i$ is $vik$.

$$\text{ICF}(\boldsymbol{X}) = \{i \mid \text{p\_value}(F(\{Y_{vi1}, Y_{vi2}, ..., Y_{vim}\})) < F_{thresh}\} \tag{2}$$

In the Mutual Info test, we select features whose ratio between the average mutual info score in the categorical case and the average mutual info score in the numerical case is higher than a threshold, where the mutual info score is computed between the feature, the other features and the target value, as shown in Equation 3, where $\bar{X}^i$ refers to the set obtained from features $X$ with the removal of the feature at index $i$ and $X_c^i$ is the categorical encoding of feature $i$.

$$\text{ICF}(\boldsymbol{X}) = \{i \mid \text{MI}(X_c^i, \{\bar{\boldsymbol{X}}^i, Y\}) / \text{MI}(\boldsymbol{X}^i, \{\bar{\boldsymbol{X}}^i, Y\}) > MI_{thresh}\} \tag{3}$$

### 3.3 FOURIER EMBEDDINGS

The other component of our preproessing method is adapting LFF developed in other fields of DL (Tancik et al., 2020; Li & Pathak, 2021; Yang et al., 2022) for overcoming the bias of deep neural networks towards overly-smooth solutions. We believe our approach to bear some simillarity with the Periodic Activation Functions proposed recently in tabular deep learning by Gorishniy et al. (2022). Differently from them, we extract two types of Fourier Embeddings: Conv1x1LFF and LinearLFF. As opposed to Periodic Activation Functions, where features are embedded separately, LinearLFF allows mixing features through a linear projection, while Conv1x1LFF extracts embeddings using parameter sharing. In Conv1x1LFF the parameters are shared by using a trainable 1D convolution over the input sequence along the number of features dimension, while in LinearLFF we pass the input sequence through a learned linear layer. The result is an output $\boldsymbol{Z}$ of shape $D \times M$, with $M$ the size of the Fourier embedding. The embeddings of size $M$ are obtained using Equation 4, where $\oplus$ is the concatenation operator.

$$\text{LFF}(\boldsymbol{Z}) = cos(\pi \cdot \boldsymbol{Z}) \oplus sin(\pi \cdot \boldsymbol{Z}) \tag{4}$$

The parameters are randomly initialized with a Gaussian distribution centred in zero, varying the number of learned frequencies through the $M$ parameter. We refer to the resulting models **ResNet+F** and **MLP+F**. Importantly, in our experiments, whether the model is using CFD or LFF is a hyperparameter in the random search space. In the figures and results that follow, we name the models that resulted from this model selection procedure **ResNet+F|C** and **MLP+F|C**.

### 3.4 BACKBONE MODELS

Our two backbone models are a simple MLP and 1D Convolutional RESNET, referred to as RESNET throughout the paper. The motivation for the choice of these particular classes of models stems from the observations of Grinsztajn et al. (2022) and Ng (2004) before them, establishing that tabular data tends to lie in a natural base and that models which are invariant to rotation, such as the MLP, suffer from a higher sample complexity in the presence of noise. Therefore, we study the effect of our feature preprocessing methods on a model that is rotationally invariant, MLP, and one that is not rotationally invariant through the use of 1D convolutions, RESNET.

We base our RESNET implementation on Hong et al. (2020). For an input of shape of shape ($N$, $F$, $D$), wher $N$ is the number of items in the batch, $F$ is the number of features and $D$ is the depth of the data, the model uses 1D convolutions of shape ($K$, $D$) along the F dimension. For the raw input, $D$ is equal with 1. As described in 3.2 and 3.3, our preprocessing results in a data depth $D$ that is either the highest number of bins in the implicitly categorical features setup and the embedding size $M$ in the LFF setup. We use a fraction $\phi$ of the feature size $F$ as the kernel size $K$, with $\phi \in [0, 1]$. The architecture consists of stacking residual blocks, each with two convolutional layers, batch normalisation, dropout and ReLU nonlinearities as well as shortcut connections from the block input to the output. The results of the final block are averaged across the last dimension and connected to a linear layer.

## 4 EXPERIMENTAL SETUP

Our experimental setup closely mirrors the setup introduced in Grinsztajn et al. (2022), including dataset processing, the multi-fold split based on dataset size, as well as train, validation and test subset sampling using the same random seeds. The benchmark consists of binary classification and regression tasks, with numerical and categorical features. The datasets are preprocessed with several schemes, such that one dataset can be part of multiple tasks, for example with numerical only or a mixture of numerical and categorical features. The datasets are subject to specific processing decisions, such as truncation to 10k or 50k samples, missing data removal, binarisation of the target value for the datasets in the classification tasks, removal of high-cardinality categorical features and low-cardinality numerical features. The datasets are split in classification and regression tasks, with subtasks containing numerical-only and heterogeneous data features. To follow the original notation, we refer to the subtasks as *numerical* and *categorical*, where the latter contains datasets with hybrid data types. Additionally, the benchmark doubles the nubmer of tasks by using "medium" and "large" subsets. In this work, we combine the medium and large sized datasets in a task, due to the number of large datasets being much smaller. The datasets in the categorical tasks, which contain a mixture of categorical and numerical features, are annotated with the indices of the categorical features, which we specifically use in combination with the base models, when we don't use our proposed preprocessing methods.

We conduct a comparative analysis on the following methods: XGBOOST, which is the reported best performing method for the benchmark, the base models MLP and RESNET, as well as their combination with implicitly categorical features binning and LFF embeddings, **MLP+F|C** and **ResNet+F|C**. We set a maximum number of 400 epochs, with checkpointing from early stopping with a patience of 40 epochs. For each model, we run 150 random seeds over the hyperparameter space. In the case of **ResNet+F|C** and **MLP+F|C**, we randomly choose between **ResNet+F** and **ResNet+C**, or between **MLP+F** and **MLP+C** respectively at each run, such that each component is used in approximately half of the runs. A complete description of the parameter spaces that we considered is presented in Appendix A.2. In order to follow the setup of Grinsztajn et al. (2022) as closely as possible, we keep the hyperparameter deciding whether or not the target variable should undergo Gaussian transformation in the regression tasks. The benchmark contains predefined splits for $k$-fold cross-validation, where $k$ is selected based on dataset size. A separate subset of the vali-

dation split is used for early stopping, with the rest of it for hyperparameter selection. We compare the test performance as accuracy or r2 score of the best hyperparameter setting run on the validation split, using accuracy for classification and Mean Absolute Error (MAE) for regression as criterion.

# 5 RESULTS

We exclude datasets where no model achieves a performance higher than 0.1, which is exactly one dataset in the regression task (yprop_4_1). We note that for this dataset, the performance is close to zero for all models and we decided to remove it in order to avoid introducing misleading statistics in the average of normalised scores. We report the average normalised performance by budget using 15 random search simulations across the 150 runs. Our experiments combine the results of 51000 runs of several models, with random hyperparameter search.

We organise our experimental results as follows: we first highlight the impact of our proposed implicitly categorical features detection and LFF embeddings by showing significant improvements for both backbone models MLP and RESNET and comparing their performances with XGBOOST and showing that the **ResNet+F|C** is a competitive model in 5.1; we then analyse the performance profiles of our compared methods in 5.2, providing a complementary picture to the aggregated performance by budget analysis; furthermore, we take a closer look of the performance gaps between the top seeds of each model in 5.3 and 5.4, highlighting that our proposed preprocessing method uncovers proper data encodings across the search budget that significantly improve over the base models, which we observe in the form of "spiking" performance. The performance landscape of the models equipped with the proposed preprocessing, in comparison with the base models empirically prove the presence of implicitly categorical features and the sensitivity of deep learning methods to such features, as well as the benefit modelling functions with more flexible decision boundaries through LFF embeddings.

## 5.1 PERFORMANCE BY BUDGET

We report the averaged normalised performance by budget across 15 random permutations of the set of random search runs in Figure 2. We normalise each score using the maximum achieved by any of the model as upper limit and the performance of a random baseline as a lower limit. For the classification tasks, due to the binary nature of the target, we set the lower bound to 0.5, while for regression we use the performance of a classifier that always predicts the mean of the target values, that corresponds to an r2 score of 0. We compare the normalised test performance of the model with the highest validation performance, where we use select by minimum validation MAE for regression and maximum validation accuracy for classification.

We first observe the significant impact of the proposed feature processing method, showcasing a consistent improvement across all tasks of **ResNet+F|C** over RESNET and **MLP+F|C** over MLP. Additionally, we observe that RESNET and MLP plateaus rather fast and significantly lag behind XGBOOST across all tasks.

Furthermore, we observe that **MLP+F|C** and **ResNet+F|C** substantially outperforms XGBOOST on both the classification tasks. On the regression tasks, **ResNet+F|C** slightly outperforms XGBOOST on the numerical datasets but not on the categorical datasets, while **MLP+F|C** still lags behind. For the regression categorical task, even though our proposed feature processing improves over the performance of the RESNET baseline, it still slightly lags behind XGBOOST on average.

As observed before, we also note that XGBOOST rapidly achieves high performances during the random search, as a less sensitive method to hyperparameter tuning which further highlights its out of the box capabilities on tabular data.

## 5.2 PERFORMANCE PROFILES

Budget plots illustrate how likely it is to find a good model given some computational budget for each algorithm, however they can hide the relative strengths of compared methods through aggregation. Furthermore they are susceptible to inflated results when some models achieve very large scores compared to the second best.

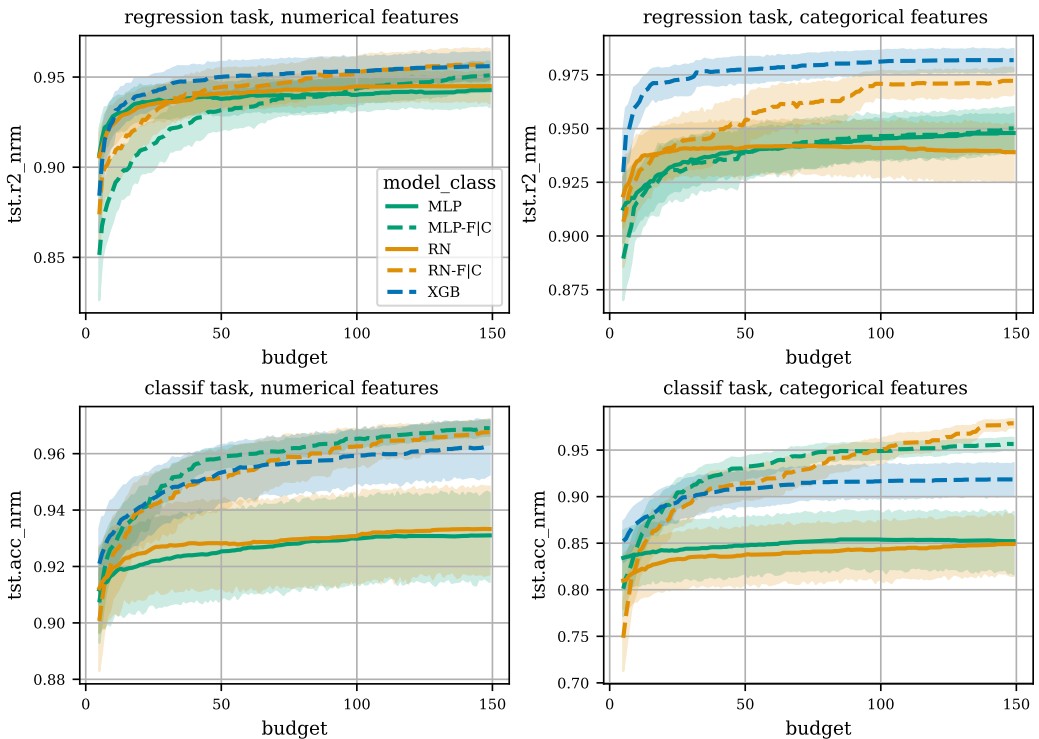

Figure 2: Performance by budget across 15 random simulations of hyperparameter optimization with 150 random seeds for each baseline.

To provide a complementary picture we use *performance profiles* (Dolan & Moré, 2002; Agarwal et al., 2021). The aim is to determine how frequently a certain method is within some distance of the best performing algorithm. Following Agarwal et al. (2021), for each normalized score $\tau$ we compute the fraction of datasets on which the performance is larger than $\tau$.

To avoid the maximisation bias that occurs when using only the best performing run found by the random search, we assume the top performing runs have been drawn independently. Alternatively we would be required to run the random search multiple times which is infeasible. We select the best performing eight runs, which for most datasets and models corresponds to the 95th percentile.

Looking at Fig. 3 it appears that XGBOOST retains much of its advantages in regression and classification tasks with categorical features across much of the performance spectrum but lines tend to overlap near $\tau = 0.97$ suggesting the top models behave similarly. These findings seem in contradiction with Fig. 2, where **ResNet+F|C** dominates the search. In the next section we look deeper into these results and clariy the apparent contradiction.

### 5.3 PROPER FEATURE ENCODING IS CRITICAL

We take a closer look at the performance landscape of the top 8 runs for each models, which corresponds approximately to the 95th percentile for each task. We take a selection of datasets with the highest normalised gap between the top runs of **ResNet+F|C** and the top runs of XGBOOST and plot the normalised scores in Fig. 5.

Looking at the first column of the **ResNet+F|C** heatmap we notice several datasets (eye_movements, year, rl, nyc-taxi-green-dec-2016) for which the top performing model has a significantly larger performance than any of the other 95th percentile runs of that model and also larger than any of the XGBOOST runs. In contrast, we notice that XGBOOST has a much higher consistency across its top performing runs.

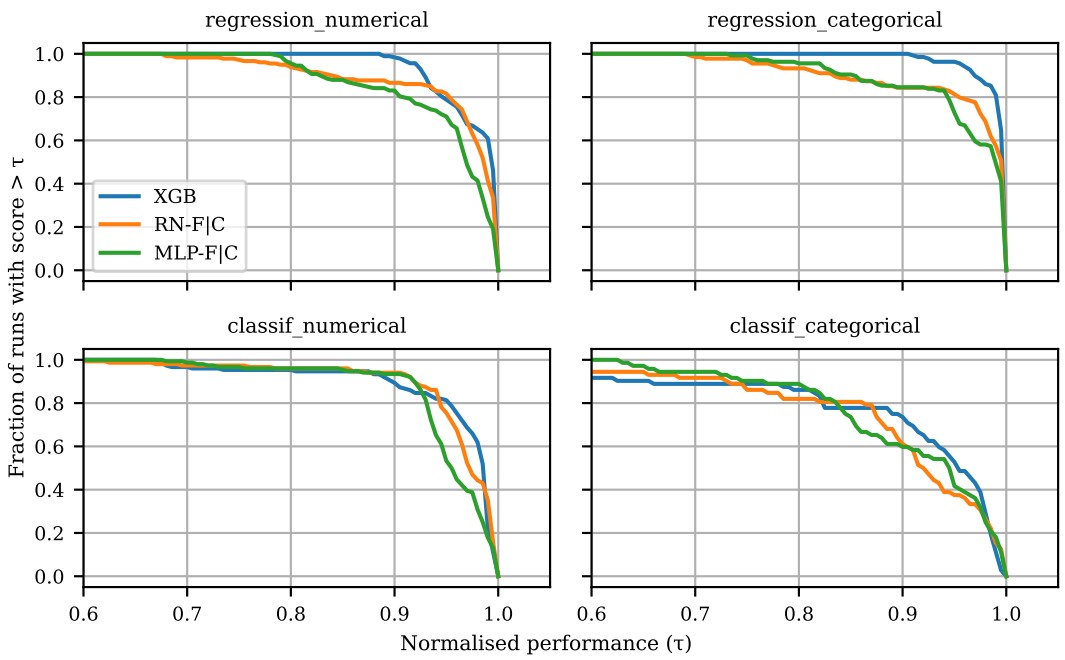

Figure 3: Performance plot on the top eight runs (approx. the 95th percentile for most tasks).

We call this observation "spiking" and attribute it to one of the statistical tests we use for implicitly categorical features detection during the hyper-parameter random search finding a good encoding of the features that can then be leveraged by the neural network. Furthermore, the large performance gaps created when our method occasionally latches on the implicit categorical features also explains the performance gaps we see in the budget plots which are dominated by the best performing training run.(Fig. 2).

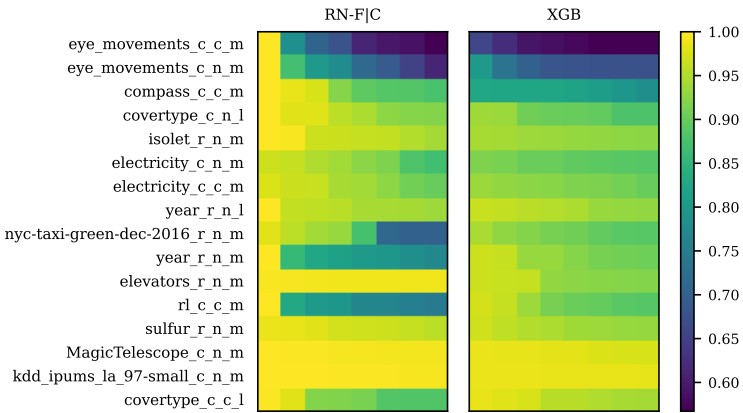

Figure 4: Heatmap of the top eight runs (corresponding to the 95th percentile for most tasks) **ResNet+F|C** and XGBOOST. We observe runs where **ResNet+F|C** achieves a performance "spike", while XGBOOST has a more consistent performance landscape

## 5.4 ABLATION: WHERE DOES THE PERFORMANCE BOOST OCCUR

We further analyse the top performing runs of **ResNet+F|C** by decoupling the **ResNet+F** and **ResNet+C** runs, in comparison with the base model RESNET. We present a performance heatmap of the top 8 runs of these models for a selection of datasets with the highest gap between the best

and the runner-up model between RESNET, **ResNet+F** and **ResNet+C**. First, we notice that the base model RESNET (RN) achieves a much more consistent performance across its top runs and it consistently lags behind **ResNet+F|C**. Additionally, while we observe that the "spiking" behavior is more frequent for **ResNet+C**, which could correspond to a proper identification of implicitly categorical features in the random search, we notice the complementarity of the two components. As previously stated, **ResNet+F** and **ResNet+C** account for approximately half of the runs compared to RESNET, as we use them in mutually exclusive fashion.

We do not observe a particular correlation between each component and a task type. For example, **ResNet+F** outperforms **ResNet+C** on some datasets from both the regression (`year` dataset) and classification (`covertype` dataset) tasks, and, conversely, datasets from both regression (nyc-taxi-green-dec-2016) and classification (`eye_movements`, `rl`, `electricity`) tasks where **ResNet+C** outperforms **ResNet+F**. These results highlight the presence of implicitly categorical features in multiple datasets, as well as datasets where the bias towards overly-smooth solutions of deep learning models hinders their performance, through the different performance landscapes of the two components and their alternating role in closing the gap to tree-based methods. Additionally, a main takeaway is that, given a search budget, simple implicitly categorical features methods succeed in uncovering the proper categorical embeddings of data that allow deep-based methods to achieve significantly higher performance margins over the base model.

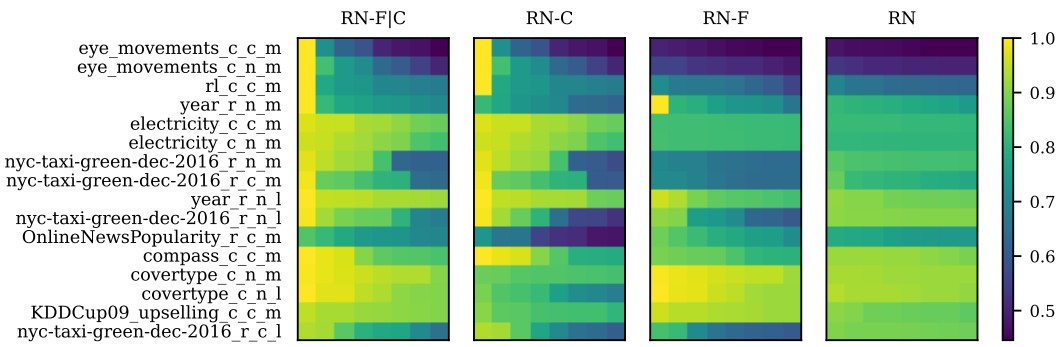

Figure 5: Heatmap of the top eight runs (corresponding to the 95th percentile for most tasks) for the base model RESNET (RN) and **ResNet+F|C** (RN-F | C). We additionally show the separated runs of **ResNet+F|C** into **ResNet+F** (RN-F) and **ResNet+C** (RN-C)

## 6  DISCUSSION AND CONCLUSIONS

In this work we report on the prevalence of implicitly categorical features in tabular data and highlight the significant performance penalty on DL methods when not addressed correspondingly when compared to tree-based methods. For addressing this newfound peculiarity of tabular data we introduce ICF which uses basic statistical methods to identify implicitly categorical features and encode them accordingly.

Complementarily, we leverage on previously introduced observations that emphasize the bias towards overly-smooth solutions as an undesirable property of models working with tabular data. Specifically we adapt LFF for tabular data applications, allowing the model to represent non-smooth decision boundaries, which are more encompassing with regard to the nature of tabular data. We employ the two proposed feature processing methods in combination with two deep learning backbone models, MLP and RESNET and show their significant performance boost over the base models and their competitiveness with DT in an extensive experimental setup,

The extensive analysis reveals examples of datasets where one of these two preprocessing methods demonstrates a significant performance advantage over the other. However, we have only scratched the surface of this phenomenon, and further analysis would be interesting as further work, such as a more detailed investigantion on the datasets particularities where these components thrive. Additionally, we hope that the present work will inspire more advanced methods of identifying implicitly categorical features or models designed to be robust to this phenomenon.

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

Table 1: Impact of categorical encoding for XGBOOST

| Model | Regression num | Regression cat | Classif num | Classif cat |
|---|---|---|---|---|
| XGBOOST | 0.7558 | 0.7770 | 0.8298 | 0.7872 |
| XGBOOST with ICF | 0.7502 | 0.7751 | 0.8271 | 0.7838 |

# A APPENDIX

## A.1 IMPACT OF ICF FOR XGBOOST

We investigate the effect of ICF feature binning for XGBOOST and we don't observe any improvement but a small decrease in performance. We report the unnormalized test performance average of the best performing hyperparameter settings across datasets and 150 random seeds in Table 1. We hypothesize that XGBOOST and DT-based methods in general inherently uncovers discontinuities introduced by implicitly categorical features and an explicit encoding of such features is unnecessary.

## A.2 HYPERPARAMETER RANGES

We present the hyperparameter spaces for the models in Tables 2, 3, 4, as well as the parameter space for the implicitly categorical features detection algorithms in Table 6 and the paramter space for the optimizers in Table 5.

Table 2: XGBOOST parameter space

| Parameter | Range |
|---|---|
| eta | logUniform [1e-05, 0.7] |
| gamma | logUniform [1e-08, 7] |
| max depth | uniformInt [3, 11] |
| subsample | uniformInt [0.5, 1] |
| lambda | logUniform [1, 4] |
| alpha | logUniform [1e-08, 1e2] |
| min child weight | logUniformInt [1,100] |
| colsample bytree | uniform [0.5,1] |
| colsample bylevel | uniform [0.5,1] |

## A.3 PSEUDOCODE

We present the pseudocode for the proposed feature preprocessing in Algorithm 1.

Table 3: MLP parameter space

| Parameter | Range |
| --- | --- |
| depth | uniformInt [2, 8] |
| width | choice [128, 256, 512, 1024] |
| activation | choice [ReLU, LeakyReLU] |
| batch normalization | choice [True, False] |
| dropout | choice [0.0, 0.5, 0.6, 0.7, 0.8, 0.9] |

Table 4: RESNET parameter space

| Parameter | Range |
| --- | --- |
| num block | uniformInt [1,3] |
| num linear | uniformInt [1,3] |
| use norm | choice [True, False] |
| norm type | choice [batch, layer] |
| use do | choice [True, False] |
| do prob | choice [0.1, 0.2, 0.3, 0.4, 0.5, 0.6] |
| downsample gap | choice [0, 1, 2] |
| increasefilter gap | choice [0, 1, 2] |
| pooling function | choice [MaxPooling, AvgPooling] |
| kernel size | uniform [0, 1] |
| activation fn | choice [ReLU, LeakyReLU] |
| lff dim: | choice [32, 64, 128, 256] |
| emb type: | choice [Conv1x1LFF, LinearLFF] |

Table 5: Optimizer parameter space

| Parameter | Range |
| --- | --- |
| opt name | AdamW |
| learning rate | uniform [0.001, 0.1] |
| eps | uniform [1e-08, 1e-04] |
| weight decay | uniform [0.0001, 0.6] |
| scheduler | CosineAnnealingWarmRestart |
| T 0 | choice [10, 20, 30, 50, 75, 100] |
| T mult | choice [1, 2] |

Table 6: ICF identification tests

| Parameter | Range |
| --- | --- |
| chi thresh | uniform [1e-50, 1e-03] |
| ANOVA thresh | uniform [1e-30, 1e-03] |
| mi thresh | uniform [0.75, 1.50] |
| min cardinality | choice [0, 10, 100] |
| max cardinality | choice [300, 500, 1000, 1500, 5000] |

---

**Algorithm 1:** Feature preprocessing

---

**Data:** Train data: input feaures $X$, targets $y$
**Result:** Output result
**Function** `ICF_test` (*X, y*, test_fn, *p_thresh*) **:**
    cat_idxs = []
    $N = X$.shape[1]
    **for** $i \leftarrow 1$ **to** $N$ **do**
        // Compute the p_value for feature $i$
        p_value = test_fn (X[:,i], X[:,:i-1] + X[i+1:,:], y)
        **if** *p_value* $<$ *p_thresh* **then**
            add $i$ to cat_idxs;
    **return** cat_idxs
**Function** `Preprocess` (*X, y*) **:**
    **select** test_fn $\leftarrow$ [chi, ANOVA, MI] ;        // Select a test function
    **select** $p\_thresh \leftarrow [p\_min, p\_max]$ ;        // Select a p_value threshold
    cat_idxs $\leftarrow$ `ICF_test` (*X, y*, test_fn, *p_thresh*) ;
    num_idxs $\leftarrow \{$i for i not in cat_idxs $\}$
    $\Phi_{cat} \leftarrow$ OneHotEncoding($X[:,$ cat_idxs$])$
    $\Phi_{num} \leftarrow$ LFF($X[:,$ num_idxs$])$
    **return** $concatenate(\Phi_{cat}, \Phi_{num})$

---

