# OpenReview forum: "Closing the gap on tabular data with Fourier and Implicit Categorical Features"
_ICLR.cc/2024/Conference — Submitted to ICLR 2024_

### Official Review · Reviewer_d92g · 2023-10-17

**Soundness:** 4 excellent
**Presentation:** 3 good
**Contribution:** 3 good
**Rating:** 5
**Confidence:** 5

**Summary:**

The paper discusses a method for deep learning on tabular data. The essence of the method is to design novel types of trainable embeddings for the data at hand, that can be. then presented to a deep net.

We have two types of propositions:

1. Categorical embeddings which are derived by comparing statistics of the data to some reference/threshold value.
2. Fourier embeddings, which are obtained through a dense or 1X1 conv. linear and then application of a Fourier transform.

The scheme is end-to-end-trainable.

**Strengths:**

A novel method that is very useful and probably impactful in the field.
A sound approach that is well motivated and with correct derivations.
An experimental evaluation that is convincing, as it entails many datasets and comparisons to logical benchmarks methods.

**Weaknesses:**

Some further discussion on related work, e.g. using Transformers for tabular data, is certainly needed. This is a must for acceptance, even after the rebuttal/revision phase.

**Questions:**

How does your method compare to Transformer-based methods for tabular data?

---

> ### Author Response · Authors · 2023-11-23
>
> Thank you for your insightful comments! We have revised our document and provided a top-level comment with more details of interest.
>
> Regarding the comparison with transformer architectures, our primary interest is not in achieving SOTA results on small datasets for tabular data, but rather in understanding and identifying the exact causes that keeps regular neural network approaches under the tree based models performance, for this specific usecase. We don't use Transformers as baselines, because we underline causes that are independent from the transformer insights, like ICF and LFF. Nevertheless, we have included related work on Transformers for tabular data in the document. We leave the comparison with transformer-based architecture as future work.

---

### Official Review · Reviewer_tmhZ · 2023-10-28

**Soundness:** 2 fair
**Presentation:** 1 poor
**Contribution:** 2 fair
**Rating:** 3
**Confidence:** 5

**Summary:**

**The paper focuses** on classification and regression problems on tabular data.

**The paper proposes** two modules for handling continuous features before passing them to a *custom convolutional ResNet-like backbone*:
- ICF (Implicit Categorical Features):  a technique for identifying "implicit categorical features" (the term introduced in the paper)
- LFF (Learned Fourier Features): an adaptation of Fourier Features ("Fourier Features Let Networks Learn High Frequency Functions in Low Dimensional Domains" by Tancik et al.)

**The main claim:** *"Our proposed feature processing and method achieves a performance that closely matches or surpasses XGBoost on a comprehensive tabular data benchmark."*

**Strengths:**

- The research direction is important: for tabular deep learning methods, it is crucial to properly handle continuous features.
- The concept of "Implicitly Categorical Features" is potentially interesting.
- The size of the used benchmark is a strength.
- I also appreciate that experiments were run with multiple random seeds.

**Weaknesses:**

Unfortunately, the paper has several major issues.

**(1) The key related work `[1]` is not properly positioned and compared against.** Unfortunately, this issue is serious:
- this submission and `[1]` have the same call for action: "let's improve how continuous features are handled by embedding/encoding them". The specific wording and informal perspective are different (in this submission, the "implicit categorical feature" perspective is suggested, while in `[1]` the motivation is more "empirical"), however, I see it as a stylistic difference: the call for action and the explored schemes are the same in nature.
- the "ICF" methods and piecewise-linear encoding from `[1]` are formally different methods, however, on the low technical level, both aim to identify a chance to discretize a given continuous feature;
- the LFF method is very similar (identical?) to either "Periodic" module from `[1]` or to Fourier Features themselves `[2]`.

To sum up, from my perspective, the key related work is not identified as such, the methods from this submission are similar to those from that related work, and the existing methods are not included as baselines.

**(2) The paper introduces a custom non-trivial architecture which seems to be unrelated to the main topic of the paper.** Specifically, in the second paragraph of Section 3, a custom architecture is introduced without any further discussion (also, to me, the usage of *convolutions* on tabular data problems is not intuitive and deserves its own discussion, or even its own paper). Unfortunately, it makes it hard to understand what is the core factor behind the reported results: the new architecture, the proposed methods or a combination thereof. I highly recommend to test the proposed methods on *existing* backbones (e.g. MLP and Transformer from `[3]`).

**(3) Regarding the presentation,** I should admit that I understand the proposed methods and the backbone to the extent that is enough to write this review, but not enough to actually implement the methods and explain them to others.

**(4) Experiments & Methodology.**
- no baselines beyond MLP and XGBoosts
- some of the datasets are represented multiple times in different forms. It makes the benchmark biased towards the repeated datasets and the notion of "the number of datasets" becomes less obvious.
- etc.

To be more specific, let's consider Figure 1:
- the figure presents 16 results, however, after removing duplicated datasets, only 11 results are left.
- In the light of `[1]`, there are no performance gaps on the following datasets: covertype, eye movements, house 16H (also, the "eye movements" dataset contains a leak and is solvable with almost perfect accuracy; it is not a well known fact though).
- On three (OnlineNews, wine quality, nyc-taxi) of the remaining 8 datasets, XGBoost wins.
- Error bars are not presented, so it is hard to judge how significant the remaining 5 wins are.
- Baselines are missing, so it can be the case that the remaining wins can be achieved with prior methods.

**References**

- `[1]` "On Embeddings for Numerical Features in Tabular Deep Learning" by Gorishniy et al.
- `[2]` "Fourier Features Let Networks Learn High Frequency Functions in Low Dimensional Domains" by Tancik et al.
- `[3]` "Revisiting Deep Learning Models for Tabular Data" by Gorishniy et al.

**Questions:**

-

---

> ### Author Response · Authors · 2023-11-23
>
> Thank you for the detailed review. Many of your observations had an impact in how we changed the presentation of our results and even some claims.
>
> (1) _The key related work [1] is not properly positioned and compared against._
>
> On a closer reading of the reference you mentioned we agree that our adaption of Learned Fourier Features is similar to Periodic Activation Functions. There still are some differences given we made some different decisions when adapting the work of Tancik et. al (our common source of inspiration). We updated the manuscript to reflect this and dropped LFF for tabular data from the main contribution list as a result.
>
> ---
>
> (2) _The paper introduces a custom non-trivial architecture which seems to be unrelated to the main topic of the paper._
>
> We give a detailed description of the ResNet 1D we used as a base model in the revision. In addition we add results using our method in top of an MLP base model as you suggested. Indeed, the proposed methods improve both the MLP and the ResNet models.
>
> ---
>
> (3) _Regarding the presentation, I should admit that I understand the proposed methods and the backbone to the extent that is enough to write this review, but not enough to actually implement the methods and explain them to others._
>
> Please check the improvements we made through this revision in the top-level comment. The addition of Fig.1 and many of the additional paragraphs on the model should give a better understanding of the overall proposal. In conjunction with the hyper-parameter ranges in the supplementary there should be sufficient details for reproducing at least parts of this work.
>
> ---
>
> (4) _Experiments & Methodology._
>
> Most of the revision focused on how to present the results better and more transparently and on relating them to our claims. Hopefully the effort of the rewrite addresses some of the concerns highlighted by your review.

---

### Official Review · Reviewer_AFsk · 2023-10-31

**Soundness:** 3 good
**Presentation:** 3 good
**Contribution:** 2 fair
**Rating:** 5
**Confidence:** 4

**Summary:**

This paper introduces two feature processing techniques - implicitly categorical feature identification and Fourier embeddings for tabular data.

**Strengths:**

- The paper is well presented.
- Evaluations are intensive/solid with good analysis.

**Weaknesses:**

- There's is limited technical contribution in the paper.
- The method here is more like feature engineering work that requires a lot of tuning to perform well.
- The performance improvement is marginal compared to XGBOOST.

**Questions:**

- Did the method use all data or only train data for categorical feature identification?
- Does the model take into account any bias in train/test distribution?
- There are some heuristic methods [1] that also discretise numerical features using, e.g., log transformation/binning rare features into one feature. It is interesting to see how such naive method compared to the proposed method.

[1] Juan, Yuchin, et al. "Field-aware factorization machines for CTR prediction." Proceedings of the 10th ACM conference on recommender systems. 2016.

---

> ### Author Response · Authors · 2023-11-23
>
> Thank you for your insightful comments! We hope to have clarified some of the aspects of our work in the top-level comment. We add our answers below:
>
> **Q1.** Did the method use all data or only train data for categorical feature identification?
>
> **A1.** We consider that we don't have access to the test data, so we only used the training samples (without validation or test).
>
> **Q2.** Does the model take into account any bias in train/test distribution?
>
> **A2.** In this work we do not account for any biases specific to a dataset. But we test over a lot of different datasets (hopefully with a different bias), in order to diminish their potential impact in the final results.
>
> **Q3.** There are some heuristic methods [1] that also discretise numerical features using, e.g., log transformation/binning rare features into one feature. It is interesting to see how such naive method compared to the proposed method.
>
> **A3.** Thank you for pointing it out, seems quite interesting and we agree that it would be an interesting comparison with our proposed method, which we will take into consideration for potential upcoming revisions of our work.
>
> We add some comments on the mentioned weaknesses below:
>
> **W1.** "Limited technical contribution"
>
> **W2.** "more like feature engineering that requires a lot of tuning"
>
> **W1-2A**. While this might be the initial feeling, we specifically identify 2 methods for closing the gap between tree based models and regular neural networks, for 3 out of 4 kinds of tasks. We empirically show their relative contribution in the ablation in section 5.4. We want to emphasize that the problem needs further exploration in order to determine what exactly causes the gap (is it the architecture biases, is it the feature preprocessing, is it the feature kind for those tabular problems, or something else).
>
> **W3**. "The performance improvement is marginal compared to XGBOOST"
>
> **W3A.** Rather than achieving SOTA on small datasets for tabular data, our primary interest lies in understanding and identifying the causes that keeps regular neural network approaches under the tree based models performance, for this specific use case. Please see the top level answer for more details.

---

### Official Review · Reviewer_xxW1 · 2023-10-31

**Soundness:** 1 poor
**Presentation:** 2 fair
**Contribution:** 1 poor
**Rating:** 3
**Confidence:** 4

**Summary:**

The authors consider the problem of the gap in performance of deep learning methods on tabular datasets compared to the best non-DL methods, i.e., tree-ensembles like xgboost.

They theorize that two key aspects not fully explored previously should be addressed: implicit categorical features, and heterogeneity of the data when it comes to tabular data.

They propose an approach consisting of 1D convolutional neural nets with residual connections, in combination with identifying implicit categorical features and encoding them, and including learned fourier features.

They experimentally evaluate the proposed approach on the same set of datasets and experimental setup as a prior work and demonstrate significant improvement with the proposed neural net approach, showing competitive or even better performance than the other DL methods and best non-DL methods

**Strengths:**

-The work is introduced very well - the motivation is clear and the existing work is very nicely summarized.

-The authors show impressive results with the proposed method - showing improved accuracy / R2 score on average across many datasets

**Weaknesses:**

-None of the methods are defined or explained in enough detail to understand what is being done and reproduce the results.  I.e., the particular model used and how it is applied, the fourier features, and the implicit categorical feature selection - none of these are clearly explained or described in enough detail to reproduce what was done.  Furthermore, what is described does not make much sense the way it is currently described.  As one example, it's stated at the beginning of modeling, a 1D CNN resnet is used - but not how this is applied to tabular data (which is not typical), what the architecture looks like or math equations, etc.  A machine learning practitioner would typically not think of applying convolution to tabular data, and the "how" and "why" of this is never explained.

-There doesn't seem to be much novelty.  Most of the proposed elements were used in prior work, including learned fourier features, resnets, and identifying implicit categorical features (indeed Grinsztajn et al., 2022 included identifying implicit categorical features as part of their data processing pipeline, as can be seen in their publicly-released code as well). The exact same dataset and experiment setup is taken from prior work and it seems like this particular method is just plugged in.  Perhaps this particular way of applying 1D convolution has not been tried with tabular data before - I can't tell because it's not completely clear what exactly was done.  Even so the novelty is very limited.

-There is no rational or explanation given for why something was done or why it might work / did work, and most of the proposed methods don't intuitively make sense.  E.g., why is chi-square test used to determine if a feature is implicitly categorical or not?  What is the rationale?  How does this make sense?  Similar how does it make sense to use 1D convolution with tabular data?  How does the excessive 0-padding make sense - if every numerical feature is also 0 padded to the maximum number of categories?  What about for the fourier features.
--Similarly, there is no analyses of results or understandings given for why anything proposed worked and how it relates to the original motivating claims.

-It's not clear how the different proposed components contribute to the improved accuracy - ideally there should be ablation study and more experiments understanding the impact of the different components.
 E.f., the different components before the neural net (identifying implicit categorical features, fourier feature transforms) should be used with other models as well, and the impact of the particular CNN model should be understood based on how it performs with and without each component as well.

-Some claims are unsubstantiated.  E.g., the authors state: "Grinsztajn et al. (2022) and Ng (2004) before them, establish that tabular data tends to lie in a natural base and that models which are invariant to rotation, such as the MLP, suffer from a higher sample complexity in the presence of noise. For this reason we select a 1D convolutional Residual Network (RESNET) to be our baseline and primary architecture since 1D convolutions are not invariant to arbitrary rotations or permutations of the data."  However, they do not show or cite that 1D CNN resnets are not invariant in the same way that MLPs are.  Since MLPs are a superset of CNNs (i.e. CNNs are MLPs with a special structure) - I doubt this claim.  The work that is cited does not mention CNNs at all.

-I don't feel that using results based on random hyper parameter search is the most reasonable way to compare the different methods, as it's not what is typically done in practice.

**Questions:**

1D convolutional neural net - why is it not rotationally invariant but MLP is (MLP is a superset of CNN)?

How is the 1D convolution applied to the tabular data?

why not use the other models / mlp like before to see if the only improvement is from that, or do the proposed changes help universally?

why is chi-square test used to determine if something is categorical?

Learned fourier features and similar were already used in past work - what's new here?

ResNet + F is never defined - what is it?

"truncation of the number of samples to 10k or 50k features," must be a typo

---

> ### Comment · Reviewer_HCF3 · 2023-11-13
> **1d convolutions**
>
> I think the way the authors suggest to use 1d convolutions is to have a hidden layer of size input_features x channels. That would be not rotation invariant. It's unclear to me whether there are pooling operations, though, which would be somewhat counter-intuitive because they rely on the ordering of features. If there is not, this is basically independent resnets for each feature that are tied together in the last layer.
>
> I totally agree that the architecture is not explained in enough detail, but I think the argument that 1d convolutions are not rotation invariant is sound.

---

> ### Author Response · Authors · 2023-11-23
>
> Thank you for your insightful comments! In addition to our top level comment and revision, we add some more specific answers to the questions below:
>
> **Q1.** 1D convolutional neural net - why is it not rotationally invariant but MLP is (MLP is a superset of CNN)?
>
> **A1.** Consider the experiments in Grinsztajn et al, where the rotated inputs and the targets and MLPs do not lose performance. Doing this for ResNet changes the performance, because of the constraints of the convolutions.
>
> **Q2.** How is the 1D convolution applied to the tabular data?
>
> **A2.** We have provided additional details in section 3.4.
>
> **Q3.** Why not use the other models / mlp like before to see if the only improvement is from that, or do the proposed changes help universally?
>
> **A3.** We have also included an MLP model in our revision and we note that our preprocessing helps this class of models as well, however it still lags behind ResNet and XGBoost on two of the four tasks, namely the regression ones. We also note that MLP uses a flattened version of the input and cannot benefit from the concatenation of embedded features across a depth channel, which might result in large input layers for MLP.
>
> **Q4.** why is chi-square test used to determine if something is categorical?
>
> **A4.** We attempt to identify implicitly categorical features with several statistical tests. Chi-squared usually tests the hypothesis that two categorical variables are independent. By categorizing a numerical feature through binning, we test whether or not the feature and the target in the classification tasks are correlated. If the test obtains a low p-value (which measures the evidence against a null hypothesis that the two categorical features are correlated). By properly adjusting the threshold, we can filter out features that don't exhibit significant correlations with the target when discretized. Additionally, we use the ANOVA test for regression or the Mutual Info, which is a metric that also accounts for one feature's correlation with the other features in the dataset and not only the target.
>
> **Q5.** Learned fourier features and similar were already used in past work - what's new here?
>
> **A5.** We have adapted Fourier Features that were previously used in tasks other than tabular data. Upon closer inspection, we drew some similarities and differences with the embedding scheme used in Gorishniy et al for tabular data. Please check the revision and the top level comment for more details.
>
> **Q6**. ResNet + F is never defined - what is it?
>
> **A6.** Sorry, it was a typo. We edited the text more carefully.
>
> **Q7.** "truncation of the number of samples to 10k or 50k features," must be a typo
>
> **A7.** Thank you for pointing it out, we indeed meant 10k or 50k samples and have made a typo, we edited the text.
>
> Regarding the weaknesses, we add some comments below:
>
> **W1.** Model and methods are not explained well enought in order to reproduce the results.
>
> **A1.** Thank you for pointing this out, we realised that our model explanation was indeed lackluster and we have added Figure 1, Algorithm 1 to explain our feature processing, as well as supplementary information of the backbone models in section 3.4.
>
> **W2.** Not enough novelty
>
> **A2.** We found no mentioning in the literature on the importance of identifying implicitly categorical features. With this work, we bring it into attention and also demonstrate the large impact it has on performance of deep learning models.
>
> **W3.** Padding
>
> **A3.** We use padding in order to concatenate all features across channels. This is used by the ResNet model which slides 1D kernels across the feature dimension of the input. We have provided more details in section 3.4. We also limit the maximum number of bins for the categorical features using a hyperprameter in the random search.
>
> **W4.** the different components before the neural net (identifying implicit categorical features, fourier feature transforms) should be used with other models as well
>
> **A4.** We have combined MLP and our preprocessing in the analysis. We observe significant improvements compared to the MLP baseline, however this model lags behind ResNet and XGBoost on the regression tasks. We also note that we cannot transpose the embeddings of each feature as in the case of ResNet and instead the input is flattened.

---

### Official Review · Reviewer_HCF3 · 2023-11-01

**Soundness:** 3 good
**Presentation:** 3 good
**Contribution:** 2 fair
**Rating:** 5
**Confidence:** 4

**Summary:**

The paper introduces four techniques to overcome shortcomings of neural networks on tabular datasets, these are:
- detecting implicit categorical features
- using a channel encoding for categorical features
- using learned fourier features
- using 1d convolutions to create non-rotationally invariant neural networks

The authors find that when searching a large space of hyper-parameters including these improvements, they can outperform XGBoost on a substantial portion of the benchmark introduced by Grinsztajn.

**Strengths:**

The question the paper addresses is highly relevant, and the methods that the paper proposes seem very interesting to investigate in this context. The paper does a thorough benchmark using an established protocol and show improvements over XGBoost. The authors summarize the current literature on the topic well.

**Weaknesses:**

While the authors summarize the current literature well, the comparison between the literature and the proposed method is somewhat lacking. While it is not feasible to reimplement all the competing methods and evaluating them, at least some of them should be compared. In particular Kadra showed that simple networks can perform well, if tuned correctly, while this paper only uses a very limited search space for the baseline MLP.

The authors use a measure that I'm unfamiliar with for evaluation, 'p-range' and don't provide a reference for it. Using this and using unnormalized scores both seem non-standard, and are not in line with the measure in Grinsztajn, which uses ADTM. Other options would be critical difference diagrams or performance profiles.

The main novelty of this work is introduced in 3.2, so this aspect should be investigated in more detail. There is many choices in the detection of implicit categoricals, and not all off these are justified or even discussed. For example, it's unclear why implicitly categorical features should have low cardinality. For example a time series with discrete jumps is something that's hard to learn for an MLP and should probably qualify as implicitly categorical, but can have arbitrary many values.
I find the definitions in 3.2.2 hard to follow and possibly an illustration would help, or maybe a reasoning why these are good criteria.

The choice of the channel-wise encoding of categorical variables seems odd, in particular since it seems that one category per variable is special, since it is the category with all the continuous features associated with it, while the others are permutation invariant - except that they depend on the permutation of the other features. The paper doesn't suggest an ordering of the categorical features, so this seems a bit strange.

The paper states that the current state-of-the-art on the Grinsztajn dataset is XGBoost, but I'm not sure if that is a fair characterization. Many of the works in the related work section simply have not been evaluated on this benchmark, as far as I know. As mentioned in the introduction, McElfresh showed that the gap between tree-based models and neural models is negligible for many datasets; I feel the phrasing in the rest of the paper should reflect that, in particular the second sentence of the introduction seems to contradict this.

Minor nitpicks
The phrase "natural base" was not entirely obvious to me when first used, maybe a different phrase or explanation when first mentioned.
The Sentence defining ResNet+C is just after the definition of the Fourier features, which is a bit confusing, and ResNet+F is not defined at all. I assume that is a typo?
In the conclusion, the paper states that it reports on the prevalence of implicitly categorical features, which I think the paper does not. It would be very interesting to study the prevalence more directly, but this seems not to be included in the paper.
I am also a bit ambivalent about the last statement about computational efficiency, since the paper uses a large search space and large computational budget to find good models.

**Questions:**

Was there a reason not to include learning rate, batch size and schedules in the searches for ResNet and the MLP? These seem quite important.
How was the threshold for including the one regression dataset chosen? Since the scores are not normalized, there is no special meaning to 0.1, is there?
How are categories ordered in the multi-channel encoding of categorical variables?
Why was a 1x1 convolution resnet chosen, and how would it compare against a standard resnet? In particular, any resnet would not be permutation invariant with respect to features because of the skip connections, right? [if the number of hidden units was chosen as the number of features, which is quite smiliar to the 1d case in some sense]
Is there a reference for the p-range metric?

---

> ### Author Response · Authors · 2023-11-23
>
> Thank you kindly for your insightful comments.
>
> **Q1.** Was there a reason not to include learning rate, batch size and schedules in the searches for ResNet and the MLP? These seem quite important.
>
> **A1.** The grid search was performed, for both ResNet and MLP, with the following ranges: lr = uniform(0.001, 0.1), Adam eps = uniform(1e-08, 1e-04), weight decay = uniform(0.0001, 0.6), with AdamW. In addition, we employed a Cosine Annealing Warm Restart scheduler with values chosen from [10, 20, 30, 50, 75, 100] for T0 and [1, 2] for Tmul. All details are now in Table 5.
>
> **Q2.** How was the threshold for including the one regression dataset chosen? Since the scores are not normalized, there is no special meaning to 0.1, is there?
>
> **A2.** There was a single dataset which was only reaching scores very close to 0.0 on all models. We decided to exclude it and picked 0.1 because it provided with a large margin between the threshold and both the underperforming and the rest of the models. We further decided on a threshold of 0.1, but this didn't include any other dataset.
>
> **Q3.** How are categories ordered in the multi-channel encoding of categorical variables?
>
> **A3.** We don't assume the data is ordinal, only categorical. Please check Figure 1 in the revision and section 3.1 for more details.
>
> **Q4.** Why was a 1x1 convolution resnet chosen, and how would it compare against a standard resnet?
>
> **A4.** For the ResNet 1D Convolutional model we employed the kernel size is a hyperparameter (see Table 4) in the range [KENREL SIZE INTERVAL]. The 1x1 convolution mentioned in the paper is only for parameterising the Learned Fourier Features, where h = pi * conv1x1(x); phi = [sin(h), cons(h)], as described in 3.3 . The reason for choosing 1x1 convolutions is to avoid mixing tabular attributes when computing the new features -- each element in the output of the FF depends only on the weights and the corresponding column in the input.
>
> **Q5.** In particular, any resnet would not be permutation invariant with respect to features because of the skip connections, right? [if the number of hidden units was chosen as the number of features, which is quite smiliar to the 1d case in some sense]
>
> **A5.** Permutation invariance was not a goal of our design. The reason we picked 1D convolutions was to make sure that we break the **rotational invariance** of models such as MLPs. As explained in the paper, prior works suggests that rotational invariance in noisy settings likely increases the sample complexity of models. We give more details about this in GA3. Circling back to the question, the 1D convolutional model is clearly not permutation invariant, with or without residual connections.
>
> **Q6.** Is there a reference for the p-range metric?
>
> **A6.** Please check our top-level comment with general answers which includes a discussion about p-range.
>
> Regarding the weaknesses, we add the following comments:
>
> **W1.1** While the authors summarize the current literature well, the comparison between the literature and the proposed method is somewhat lacking. While it is not feasible to reimplement all the competing methods and evaluating them, at least some of them should be compared.
>
> **A1.1** While stronger methods than XGBoost might exist, our aim in this work was to first close the gap to a well-known and used tree-based method.
>
> **W2.** The authors use a measure that I'm unfamiliar with for evaluation, 'p-range' and don't provide a reference for it. Using this and using unnormalized scores both seem non-standard, and are not in line with the measure in Grinsztajn, which uses ADTM. Other options would be critical difference diagrams or performance profiles.
>
> **A2.** Regarding 'p-range', please see the top level comment.
>
> **W3.1** "low cardinality"
>
> **A3.1** Indeed, low cardinality does not always imply a feature is categorical. We use a hyperparameter to decide whether or not low-cardinality features should be automatically considered categorical and this decision is a result of the sensitivity of some of the statistical tests that we use to low cardinality features, which might not properly be detected. Additionally, converting high cardinality columns to categorical becomes lossy in our proposed binning with a fixed number of bins, so we set a threshold of how large the cardinalities should be.

---

### Author Response · Authors · 2023-11-23
**Major manuscript revision**

We thank the reviewers for the many suggestions, observations and questions that determined us to rewrite and extend large portions of the original manuscript. We believe the new version goes to a great length in addressing two of the key issues highlighted by reviewers:

- a lack of details and intuitions in describing the methods
- some confusion regarding the chosen metrics for reporting our results

To summarize, here are the main changes we made. For details please check the paper, we highlighted in blue all the edits.

1. Add a detailed graphic illustrating our method (Fig. 1).
2. Add a detailed description of the 1D ResNet model used in most of our experiments (Sec. 3.4)
3. Rename the _p-range_ metric to what it actually is: _performance profile_ and cite the relevant reference and why we use it. Different from the previous _p-range_ plots is that we use the top runs resulted from the random search and not only the best performing method. (Sec. 5.2 for a detailed discussion)
4. Replace the previous bar plot in Figure 1 with a heatmap of scores that allows a transparent interpretation of the performance of each model. We use this new visualisation for interpreting the ablation study as well.
5. Add MLP experiments as an alternative to the ResNet base model.
6. Provide a better discussion of the work of Gorishny et al. On a more carefull reading we agree our adaption of Learned Fourier Features for tabular data bears some similarity with Periodic Activation Functions of Gorishny, given the same source of inspiration (Tancik, etc.). There are some differences that we discuss in the text but this similarity determined us to drop the corresponding item from the "Contributions" list in the first section of the paper.
7. We add a detailed algorithm in the Supplementary.

We hope these changes clarify most of the previous concerns.

---

### Meta-Review · Area_Chair_3J8i · 2023-12-05

**Metareview:**

The goal is clear and worthy (identifying the causes that keep regular neural network approaches under the tree based models performance).
The originality of the approach (compared to  "On Embeddings for Numerical Features in Tabular Deep Learning" by Gorishniy et al.) was questioned by Rev tmhZ.
Aspects unclear/discussion missing: convolutional part; prevalence of implicit categorical features.
As put by Reviewer tmhZ, the authors did a major revision of the paper - that would require another reviewing process.

**Justification For Why Not Higher Score:**

N/A

**Justification For Why Not Lower Score:**

N/A

---

### Decision · Program_Chairs · 2024-01-16

Reject